# Changes in Body Composition During Intensive Care Unit Stay and Outcomes in Patients with Severe COVID-19 Pneumonia: A Retrospective Cohort Study

**DOI:** 10.3390/v17050643

**Published:** 2025-04-29

**Authors:** Hayato Nakabayashi, Junko Yamaguchi, Ken Takahashi, Yasuyoshi Kai, Kosaku Kinoshita

**Affiliations:** Division of Emergency and Critical Care Medicine, Department of Acute Medicine, Nihon University School of Medicine, 30-1 Oyaguchi Kamimachi, Itabashi-ku, Tokyo 173-8610, Japan; nakabayashi.hayato@nihon-u.ac.jp (H.N.); takahashi.ken08@nihon-u.ac.jp (K.T.); kai.yasuyoshi@nihon-u.ac.jp (Y.K.); kinoshita.kosaku@nihon-u.ac.jp (K.K.)

**Keywords:** body composition, ICU stay, COVID-19 pneumonia

## Abstract

This single-center retrospective observational study investigated the association between changes in body composition during hospitalization and outcomes in patients with severe coronavirus disease (COVID-19) pneumonia. Body composition was assessed using chest computed tomography (CT) within 3 days of intensive care unit admission and follow-up CT within 14 days. The study population comprised 89 adult patients, among whom 57 survived. The median APACHE II score on admission was 16. Initial CT scans showed that the iliopsoas muscle volume, sum of the pectoralis major and minor muscle areas, and erector spinae muscle (ESM) area were significantly larger in survivors than in non-survivors (*p *= 0.019, 0.011, and 0.001, respectively). Subcutaneous fat tissue (SAT) volume was higher in survivors (*p *= 0.003), and the rate of change in the SAT volume was lower in survivors (*p *= 0.043). Multivariate logistic regression analysis revealed that a high APACHE II score (OR: 0.834, 95% CI: 0.741–0.938, *p *= 0.002) and small ESM area (OR: 1.001, 95% CI: 1.000–1.002, *p *= 0.031) were independent predictors of mortality. In conclusion, the loss of supporting respiratory muscles, particularly ESM, may play a critical role beyond general acute sarcopenia, and the preserved SAT in non-survivors may reflect abnormal glucose metabolism due to severe inflammation.

## 1. Introduction

Research on coronavirus disease (COVID-19) pneumonia has provided high levels of evidence identifying older age [1], male sex [2], and physical inactivity [3] as major risk factors for disease severity. These factors align with previously reported predictors examined to clarify the relationship between skeletal muscle volume and poor outcomes in severely ill patients requiring intensive care unit (ICU) admission [4,5]. Skeletal muscle volume at admission has been shown to be associated with survival in critically ill patients [6]. Rapid muscle atrophy occurs following ICU admission in critically ill patients with general diseases, with reports indicating up to an 18% reduction in muscle volume within 10 days [7].

A recent single-center cohort study investigated the association between the cross-sectional area of the erector spinae muscle (ESM) and subcutaneous fat, measured using computed tomography (CT) scans performed within 24 h of ICU admission, and patient prognosis. The study reported that the cross-sectional area of the ESM was an independent predictor of 6-month survival in the multivariate analysis [8]. Another study identified severe obesity as a factor contributing to poor outcomes in COVID-19 pneumonia [9]. These findings suggest that body composition at admission, including muscle and fat volumes, is a potential risk factor for severe disease progression in COVID-19 pneumonia. Nonetheless, while some reports have explored the relationship between muscle and fat volumes and outcomes of patients with COVID-19 pneumonia [10,11], most are limited to post hoc analyses of small datasets [12]. In addition, some reports have indicated an association between reduced psoas muscle volume and COVID-19 severity [13,14]; however, critically ill patients account for only 4% of the study population [13]. Studies focusing on body composition assessed using CT at ICU admission in patients with COVID-19 pneumonia are limited [15,16,17], and no research has clarified the changes in body composition during treatment or their relevance to patient outcomes. Furthermore, the relationship between prognosis and changes in muscle or fat cross-sectional area or volume during hospitalization is poorly understood.

COVID-19 pneumonia is a waste disease [18], and muscle loss during hospitalization has been hypothesized to be more pronounced in patients with COVID-19 pneumonia than in those with other critical illnesses. In this single-center retrospective observational study, we aimed to investigate the association between body composition at ICU admission and patient outcomes and explore the relationship between changes in body composition during hospitalization and outcomes in critically ill patients with COVID-19 pneumonia.

## 2. Materials and Methods

### 2.1. Study Design and Patients

This single-center retrospective observational study used data from hospital databases and patients’ clinical records. Ethical approval for this study was obtained from the Institutional Review Board of Nihon University Itabashi Hospital (approval no. RK-240514-3). The requirement for informed consent was waived by the approving authorities due to the retrospective nature of this study.

The study population comprised patients aged ≥ 20 years who were admitted to the ICU at our hospital from 1 April 2020 to 30 September 2021. These patients were diagnosed with severe COVID-19 pneumonia based on positive polymerase chain reaction results for severe acute respiratory syndrome coronavirus 2 (SARS-CoV-2). Severe COVID-19 pneumonia was defined as a clinical condition requiring ICU admission for treatment or necessitating mechanical ventilation. Patients aged < 19 years, those who had already received treatment at another hospital, and post-cardiac arrest resuscitation cases were excluded from this study. The primary outcome was discharge from the ICU or transfer, and this was classified into two groups: survivors and non–survivors.

The epidemiological characteristics and general severity of critical illness were evaluated by extracting the Acute Physiology and Chronic Health Evaluation (APACHE) II score [19], an indicator of the severity of common diseases, and the Sequential Organ Failure Assessment (SOFA) score [20,21], an indicator of the severity of organ failure. In addition, the following were extracted from medical records: (1) markers used to assess COVID-19 severity, including D-dimer, ferritin, and KL-6 [22]; (2) levels of biomarkers known to be associated with COVID-19 pneumonia severity; and (3) comorbidities, which were extracted and used to calculate the Charlson Comorbidity Index (CCI) and nutritional status, which is a predictor of patient outcomes. Vital sign values, as well as blood and biochemical findings recorded at admission, were obtained. The partial pressure of arterial oxygen/fraction of inspired oxygen (P/F) ratio was calculated as an indicator of oxygenation impairment at admission. Lactate levels were also determined to evaluate shock.

Patients’ body composition was assessed by measuring the cross-sectional areas of the ESM and pectoralis major and minor muscles, the volume of the psoas major muscle, and volumes of subcutaneous and visceral fat in the abdominal region. Muscle and fat measurements were conducted according to established protocols using SYNAPSE VINCENT version 6.7.0007 (Fujifilm Medical Co., Ltd. Tokyo, Japan). Muscle and fat areas and volumes, as well as CT attenuation values, were measured using automated analysis and manual tracing (Appendix A). Two trained radiologists performed tracing after verifying the absence of measurement errors. Our ICU lacked a formal protocol; however, we adhered to appropriate nutritional guidelines based on Japanese recommendations [23,24] and international guidelines [25] for elderly patients with severe sepsis in the ICU or those with an undernutrition state related to poor outcomes. In addition, we expended efforts to provide critically ill patients with adequate energy and protein intake.

Nutritional status was evaluated using height, weight, body mass index (BMI), and controlling nutritional status (CONUT) score [26] at admission. Given that malnutrition is suspected to be associated with poor outcomes in critically ill patients [27], patients’ CONUT scores at admission were calculated as a screening tool for malnutrition. The CONUT system includes only the serum albumin level, peripheral blood lymphocyte count, and total cholesterol level in its assessment of nutritional status [28]. The total caloric and protein intakes within the first 7 days of ICU admission were calculated to assess acute-phase nutritional therapy. The use of COVID-19-specific drugs for treatment during the study period, including favipiravir, remdesivir, nafamostat, tocilizumab, baricitinib, and steroids, was also investigated. Body composition was measured using the initial chest CT scan obtained within 3 days of hospitalization and a subsequent chest or whole-body CT scan taken within 14 days of hospitalization for treatment planning.

For patients with follow-up CT scans during hospitalization, the days of interval between first and secondary scans were calculated. The rates of change in the muscle and fat areas and volumes were determined by comparing the follow-up CT images with the initial CT images. For standardization, the rate of change was divided by the interval (in days) between scans, and this value was shown as the daily rate of change for each muscle and fat parameter.

### 2.2. Statistical Analysis

Statistical analysis of the data obtained from the study population was performed using SPSS version 29 (IBM Corp., Armonk, NY, USA) and EZR version 1.67 (SAS Institute, Cary, NC, USA) [29]. Data are presented as mean ± standard deviation, median (interquartile range or minimum and maximum values), or number of cases (%), as appropriate. Statistical significance was set at *p* < 0.05.

Continuous variables were compared using Student’s *t*-test or the Mann–Whitney U test and Wilcoxon signed-rank test, as appropriate. Categorical variables were compared using the chi-squared (χ^2^) or Fisher’s exact probability test. Multiple groups were compared using the Kruskal–Wallis test. Bivariate correlations were analyzed using Spearman’s rank correlation coefficient.

In the multivariate model, variables with a *p*-value of <0.2 were considered explanatory variables [30]. Multivariate logistic regression analysis was performed to identify factors associated with mortality.

The number of variables that could be included in the model was limited due to the limited sample size. Therefore, we carefully selected variables based on objective and scientifically valid clinical reasons to avoid subjective bias as much as possible, and the explaining factor of each multicollinearity was considered.

Consequently, the APACHE II score, the volume of the psoas muscle, the total area of the pectoralis major and minor muscles, ESM area, BMI, blood glucose, D-dimer levels [µg/mL], lactate levels (mmol/L), mechanical ventilation use, and nafamostat use were transferred to the multivariate model.

## 3. Results

A total of 108 patients were admitted during the study period, of whom 19 were excluded: 3 due to cardiac arrest at admission, 2 with moderate COVID-19 pneumonia that was not severe, and 14 who had already been treated in other hospitals. Ultimately, 89 patients were included in this study (Figure 1).

Comparative analyses were conducted between the survivors at the time of discharge from the ICU or transfer (*n *= 57) and non-survivors (*n *= 32). The mean patient age was 65.4 years, with male patients accounting for 77.5% (*n *= 69). The SOFA and APACHE II scores at admission were 4 and 16, respectively. The median P/F ratio at admission was 90.5. The median length of ICU stay was 5 days. A total of 57 (64.0%) patients survived after discharge (Table 1A). Non-survivors exhibited significantly higher age, SOFA, and APACHE II scores, and ICU stay duration, than survivors. No significant difference was observed in the P/F ratio at admission. The CCI scores in the non-survival group were significantly higher than those in the survival group (Table 1A Comorbidities).

Table 1B shows the association between COVID-19-specific therapeutics and outcomes. Nafamostat use was significantly more frequent among non-survivors. In contrast, the use of other drugs did not differ significantly between the groups.

No significant difference was observed in height or weight between survivors and non-survivors; however, BMI was significantly higher in survivors. While the CONUT score in both groups indicated moderate malnutrition, no significant difference was observed. The median total caloric and protein intakes during the first 7 days of ICU admission did not differ significantly between the two groups. These results suggest no differences in the nutritional status at admission or subsequent nutritional management between survivors and non-survivors (Table 1C).

### 3.1. Body Composition at Admission and Subsequent Rate of Change.

Initial CT scans showed that the psoas muscle volume, combined pectoralis major and minor muscle areas, and ESM area were significantly greater in survivors than non-survivors (*p *= 0.019, 0.011, and 0.001, respectively). In contrast, the subcutaneous fat volume was significantly greater in survivors than in non-survivors (*p *= 0.003; Table 2).

The daily rates of change for all muscle groups did not significantly differ between the survivor and non-survivor groups. However, regarding fat volume, the daily rate of change in the subcutaneous fat volume in survivors showed a decreasing tendency, whereas non-survivors exhibited an increase in secondary CT (*p *= 0.043; Table 3).

Muscle mass is influenced by age and sex, with advanced age and male sex being recognized as risk factors for severe COVID-19 pneumonia and mortality. In the study cohort, the psoas muscle volume and ESM area showed a tendency to decline with increasing age, particularly beyond the age of 40 (Appendix A).

### 3.2. Multivariate Analysis (Logistic Regression)

The factors associated with mortality were identified using a bivariate analysis of the APACHE II score and various muscle mass measurements obtained at admission to the ICU. Variables with *p*-values < 0.2 in the bivariate analysis were included as explanatory factors in the multivariate model. Multiple logistic regression analysis was performed to calculate the odds ratios [ORs] with 95% confidence intervals [CIs].

The analysis revealed that a higher APACHE II score (OR: 0.834, 95% CI: 0.741–0.938, *p *= 0.002) and smaller ESM area (OR: 1.001, 95% CI: 1.000–1.002, *p *= 0.031) were independent predictors of mortality (Table 4). The model was as follows: Score = −1.777 × APACHE II score + 0.001 × total ESM area + 1.102.

The predictive accuracy of the model was evaluated using receiver operating characteristic (ROC) curves. The cutoff score for this model was 0.730. Using this cutoff, the area under the ROC curve (AUROC) was 0.809 (95% CI: 0.715–0.903, *p *< 0.0001), indicating good prediction accuracy (Figure 2). In addition, the Hosmer–Lemeshow test yielded a value of 0.812, confirming that the model was well calibrated to the data. The hit rate of the model was 73.5%. This result showed good accuracy as a predictive model compared with outcome prediction using the APACHE II score alone (Appendix A).

## 4. Discussion

### 4.1. Low ESM Area as an Independent Factor for Poor Outcomes

Acute skeletal muscle wasting in critically ill patients is associated with increased morbidity and mortality [27]. This study’s findings indicated that low ESM area was an independent factor for poor outcomes.

The ESM is a surrogate indicator of all skeletal muscles in the body. Heusden et al. reported that the ESM was a good surrogate indicator of total skeletal muscle mass in the psoas muscle [31]. In addition, differences in body composition between Asians, such as the Japanese, enrolled in this study, and Western populations have been suggested, making direct comparisons difficult. Ishida et al. confirmed the correlation between lumbar skeletal muscle mass and chest skeletal muscle mass in Japanese individuals, showing that total skeletal muscle mass can be predicted using chest muscle mass [32].

In this study, the ESM area, rather than the psoas muscle volume or area of any other muscle, such as the pectoralis muscle area, was found to be an independent factor related to mortality, which might be due to the functional roles of the ESM, particularly in relation to respiratory function. Respiratory function arises from the contraction and expansion of the thorax, and a reduction in muscle mass related to this activity is associated with poor outcomes in pneumonia [33,34]. The diaphragm, a major inspiratory muscle, and the abdominal muscles, which are involved in expiration, usually act when respiratory or metabolic demands increase. The ESM, which consists of the longissimus, iliocostalis, and spinalis muscles, primarily functions as an anti-gravity muscle to maintain an upright posture and stabilize the spine. However, under conditions where the respiratory demand increases, these postural muscles can function as respiratory muscles [35].

The bridging exercise is one type of strength training that increases the maximal voluntary isometric contraction (%MVIC) of the ESM. However, using a forced breathing pattern as a load reportedly increases the %MVIC equivalent to that of the bridging exercise [36]. Therefore, a low volume of the ESM area at admission is likely to increase mortality in COVID-19 pneumonia and in severe pneumonia. We believe that our results indicate the important role of the ESM in respiratory function in severe pneumonia, especially in cases of respiratory distress.

An association between reduced ESM area and poor outcomes has also been observed in community-acquired pneumonia [37]. The predictive accuracy of the model (Score = −1.777 × APACHE II score + 0.001 × total ESM area + 1.102) created using the independent factors in the logistic regression model (Table 4) indicated good prediction accuracy, similar to the AUROC curve. Specifically, the cut-off score was 0.730 and the AUROC was 0.809 (95% CI: 0.715–0.903, *p *< 0.0001). The Hosmer–Lemeshow test yielded a value of 0.812, confirming that the model was well calibrated to the data. The hit rate of the model was 73.5%.

### 4.2. Mechanisms of Muscle Loss in COVID-19 Pneumonia: Acute Sarcopenia, ICU-Acquired Weakness (ICU-AW), and Renin–Angiotensin–Aldosterone System (RAAS) Dysregulation

In the present study, the chest muscle area, ESM area, and psoas muscle volume decreased during the ICU stay, and changes in muscle mass were observed over a median of 11 days (Table 3).

This decline occurred within 28 days following acute illnesses, such as viral infections, and the mechanism of acute sarcopenia could be similar to the changes observed in this study. Acute sarcopenia refers to a decline in muscle strength and skeletal mass that occurs within 28 days of exposure to stress due to acute illness [38]. This mechanism is related to the mass reduction caused by prolonged bed rest. Bedrest leads to an imbalance between protein synthesis and degradation, resulting in skeletal muscle loss [39]. The first week of onset is associated with the most significant rate of muscle loss, with approximately 1% of the total skeletal muscle mass lost per day [40]. A similar mechanism may have occurred in the patients in this study. Beyond the mechanism of acute sarcopenia caused by bed rest, the impact of ICU-AW has been suggested. ICU-AW results from axonal neuropathy and myopathy, leading to muscle weakness and atrophy [41]. Neuromuscular dysfunction has also been reported in severe COVID-19 pneumonia cases [42]. While acute sarcopenia due to bed rest is a typical mechanism for ICU-AW, it has been suggested that in the case of COVID-19, angiotensin-converting enzyme type II (ACE II), which acts as the cellular receptor for SARS-CoV-2, could contribute to muscle mass loss through disturbances in the RAAS. ACE II depletion, induced by its role as a SARS-CoV-2 receptor, inhibits the breakdown of angiotensin II into angiotensin 1–7 (Ang1–7), leading to increased inflammation via the angiotensin type 1 receptor. Increased inflammation activates the ubiquitin–proteasome system, promoting protein degradation [43].

### 4.3. Inflammation Accelerates Suppression of Lipolysis as a Mechanism for Subcutaneous Fat Accumulation in Severe Cases

Severe obesity is a known risk factor for the progression of COVID-19 pneumonia. The degree of obesity and its association with the risk of severe COVID-19 pneumonia differs significantly across ethnic groups. Obesity risk assessment using BMI shows that the risk of severe disease increases by approximately two-fold in Western populations, whereas in Asian populations, the risk of severe disease and mortality is over five-fold [44]. Body composition is a key contributor to this difference; Asian populations tend to have a greater distribution of visceral adipose tissue (VAT) than subcutaneous adipose tissue (SAT) at the same BMI, making them more prone to metabolic disorders than Western populations [44,45]. These findings suggest that VAT is particularly associated with poor clinical outcomes among the obesity indices.

Yasuda et al. reported that VAT volume correlates with peak C-reactive protein levels during hospitalization, and patients with higher VAT had significantly higher in-hospital mortality rates. A study using a visceral obesity rat model demonstrated that SARS-CoV-2 infection increased the presence of SARS-CoV-2 positive cells and viral genomic RNA in alveolar regions, accompanied by the upregulation of viral response genes and elevated cytokine release [46].

A notable finding in this study was the paradoxical change in SAT volume: SAT volume decreased in the survival group but increased in the mortality group. Interleukin-6 (IL-6) has been reported to be a marker associated with respiratory failure in COVID-19 pneumonia, with higher IL-6 levels observed in critically ill and deceased patients [47]. IL-6 has also been implicated in insulin resistance in several studies [48]. In this study, the mortality group may have had higher IL-6 levels than the survival group, potentially leading to increased insulin resistance. Therefore, suppressed hormone-sensitive lipase activity may have inhibited fat breakdown, preventing fat accumulation from being utilized as an energy source.

Given that many study participants were elderly, it is important to note that frailty is common among older adults in Japan. In these patients, SAT serves as a critical energy reserve. The observed reduction in SAT volume in the survivor group suggests that these patients could effectively mobilize fat stores for energy during the disease course.

In this study, no significant differences in energy or protein intake over 7 days were observed between the survival and mortality groups. However, the results of our study suggest that more attention should be paid to the importance of appropriate nutritional therapy for patients with severe COVID-19 pneumonia in the ICU. Acute skeletal muscle wasting in critically ill patients is associated with increased morbidity and mortality [27], highlighting the importance of appropriate nutritional therapy for severe illnesses.

### 4.4. Limitations

This study had some limitations. First, establishing a control group was not possible, and there exists a potential for selection bias, given that this was a retrospective single-center study.

Second, we could not comprehensively assess multifactorial patient backgrounds that influence body composition, such as comorbidities and medication history, especially when examining the influence of pre-existing medications, including ACE II inhibitors, which are related to accelerated poor outcomes for COVID-19 and activities of daily living, or other comorbidities that may affect body composition. However, the effects of these factors remain unclear.

Third, the effects of in-hospital treatment on body composition have not been thoroughly investigated. In addition, the timing and extent of the rehabilitation interventions during hospitalization, which are expected to affect body composition, were not considered.

Fourth, the median age of the study population was 65 years, indicating a predominance of older patients and a lower frequency of severe cases of visceral obesity, which are more common in younger populations. This limits the generalizability of our results.

Finally, the sample size was insufficient due to the retrospective nature of this clinical observational study. There is a possibility of missing important variables. Multifactorial influences on body composition were not always thoroughly investigated, and confounding factors were not sufficiently accounted for in the analysis. Therefore, future prospective observational studies that appropriately consider confounding factors, such as age, sex, and comorbidities that affect muscle and fat mass, might provide a more detailed understanding of the relationship between body composition and clinical outcomes, which could contribute to strategies to prevent severe COVID-19 pneumonia.

## 5. Conclusions

High APACHE II score and small ESM area at admission were independent risk factors for mortality in patients with severe COVID-19 pneumonia in the ICU. The function of the ESM as an auxiliary respiratory muscle during periods of increased respiratory demand suggested that the skeletal muscle loss throughout the body, which is commonly reported in patients with severe acute sarcopenia, may not be the only factor. Furthermore, subcutaneous fat did not decrease during hospitalization in the severe group, suggesting that the decrease in utilization efficiency was associated with abnormal glucose metabolism due to severe inflammation.

## Figures and Tables

**Figure 1 viruses-17-00643-f001:**
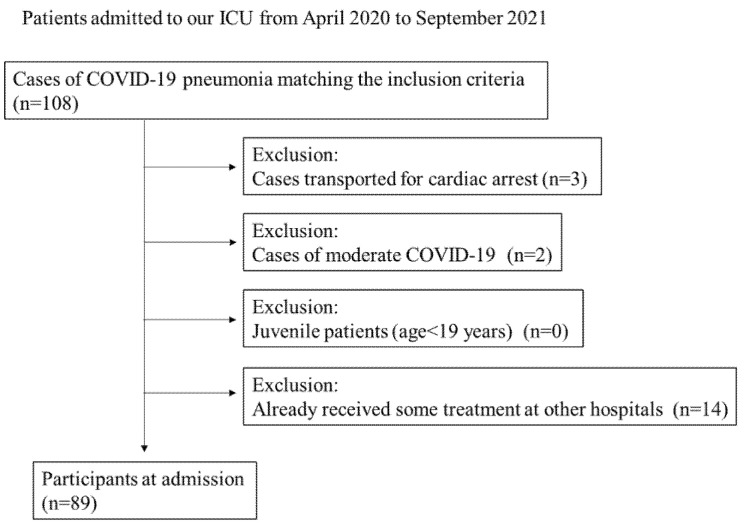
CONSORT flow diagram of patients included in the study (*n *= 89).

**Figure 2 viruses-17-00643-f002:**
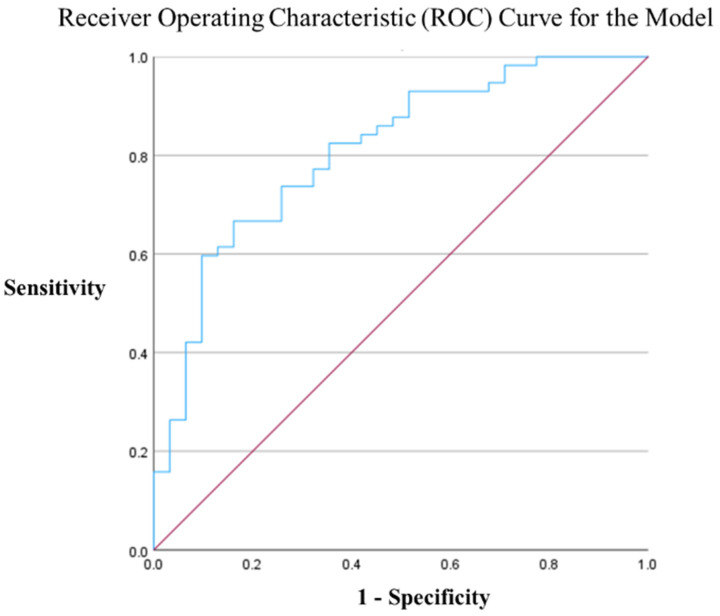
ROC curve for the predictive model combining the APACHE II score and the ESM. The AUROC value was 0.809 (95% CI: 0.715–0.903, *p *< 0.0001). The cutoff score for the prediction model was 0.730. The Hosmer–Lemeshow test yielded a value of 0.812. The hit rate of the model was 73.5%. Blue line shows ROC curve and red line shows diagonal reference line.

**Table 1 viruses-17-00643-t001:** Comparison of characteristics between survivors and non–survivors.

Factors	All (*n*= 89)	Survivors (*n* = 57)	Non-Survivors (*n* = 32)	*p*-Value *
**(A) Parameters**				
Age (years)	65.4 (57–76)	60 (54–73)	74.5 (66.5–78.3)	<0.001
Male (%)	77.5%	73.7%	84.4%	0.246
APACHE Ⅱ score	16 (12–21)	14 (11–18)	20 (16–24.3)	<0.001
SOFA score	4 (3–6)	4 (3–5)	5 (4–8)	0.040
Presence of Shock, n (%)	2:87	1:56	1:31	0.675
Lactate (mmol/L)	1.4 (1.1–2.0)	1.3 (1.0–1.9)	1.65 (1.2–2.6)	0.043
ICU stay (days)	5 (2–11)	3 (2–7)	10 (4–23)	<0.001
P/F Ratio	90.5 (65.6–134.9)	80.6 (65.6–142.5)	99 (67–132.7)	0.752
D–Dimer (μg/mL)	1.7 (1.0–7.8)	1.2 (1.0–2.7)	5.6 (1.7–11.8)	0.001
Ferritin (ng/mL)	679.9 (447.0–1406)	763.1 (467.0–1435)	629.3 (366.6–1305)	0.350
KL–6 (U/mL)	430.0 (236–740)	502.5 (290.8–817)	355 (197.5–556)	0.229
Comorbidities				
CCI [min, max]	1 [0, 9]	0 [0, 9]	2 [0, 6]	0.01
Agea-djusted CCI [min, max]	4 [0, 11]	[3, 11]	[0, 10]	0.01
Smoking History *n* (%)	41:48	23:34	18:14	0.149
Hypertension *n* (%)	47:42	30:27	17:15	0.964
Diabetes Mellitus *n* (%)	39:65	24:41	15:24	0.875
Chronic Heart Failure *n* (%)	13:76	4:53	9:23	0.012
Malignancy *n* (%)	13:76	6:51	7:25	0.146
Respiratory Disease *n* (%)	7:82	3:54	4:28	0.224
End–Stage Renal Disease *n* (%)	3:86	2:55	1:31	0.923
**(B) Treatment**				
Mechanical Ventilation *n* (%)	32:57	14:43	18:14	0.003
HFNCO, *n* (%)	30:59	22:35	8:24	0.193
drugs				
Favipiravir, *n* (%)	33:56	18:39	15:17	0.152
Remdesivir, *n* (%)	59:30	41:16	18:14	0.133
Nafamostat, n (%)	21:68	9:48	12:20	0.021
Tocilizumab, *n* (%)	55:34	33:24	22:10	0.312
Baricitinib, *n* (%)	11:78	8:49	3:29	0.522
Anticoagulants, *n* (%)	57:32	49:8	31:1	0.101
Steroids, *n* (%)	72:17	45:12	27:5	0.532
**(C) Nutrition and Glycemic status**				
Height (cm)	166.9 (159.7–170)	167.0 (159.8–171)	165.6 (159.8–170.0)	0.715
Weight (kg)	69.0 (58.5–78.0)	70.6 (58.8–80.7)	62.5 (56.8–70.4)	0.059
BMI (kg/m^2^)	24.8 (22.3–27.7)	25.5 (22.7–29.1)	24.0 (22.1–26.2)	0.038
CONUT score	7 (5–9)	7 (5–8)	7.0 (5.0–9.0)	0.493
Lymphocyte Count (cells/µL)	600 (400–1000)	600 (400–900)	700 (475–1100)	0.412
T–Cho (mg/dL)	150.5 (123.8–184.5)	151.5 (125.3–186.3)	145.0 (123.5–178.3)	0.402
Albumin (g/dL)	2.9 (2.6–3.2)	3.0 (2.6–3.3)	2.9 (2.6–3.2)	0.278
Total Caloric Intake over 7 Days (kcal)	6090 (3840–8000)	7020 (3840–8400)	5572 (4080–7375)	0.320
Total Protein Intake over 7 Days (g)	280 (161–342)	301 (150–378)	268 (182–320)	0.209
Blood Glucose (mg/dL)	162 (119–224)	152 (115–205)	204 (134–273)	0.089
HbA1c (NGSP (%))	6.5 (6.0–7.3)	6.4 (6.0–7.2)	6.6 (6.0–7.5)	0.739

Table 1 (**A**), Parameters of the patients in this study; (**B**), Treatment the patients this study received; (**C**), Nutrition and Glycemic Status of the patients this study at admission; * Categorical variables are presented as numbers (*n*) and percentages (%), and nonparametric continuous variables are presented as medians and interquartile ranges (first quartile and third quartile). Continuous variables were compared using the Mann–Whitney U test. The Chi-square tests or Fisher’s exact probability tests were performed for categorical variables. We determined the optimal cutoff points and significance level to be 5%. Abbreviations: Survivors, Survivors at the time of discharge from the ICU or transfer; Non-Survivors, Non-survivors at the time of discharge from the ICU or transfer; APACHEII score, Acute Physiology and Chronic Health Evaluation II score; SOFA score, sequential organ failure assessment; P/F ratio, Partial pressure of arterial oxygen/Fraction of inspired oxygen; KL-6, Sialylated carbohydrate antigen; CCI, Charlson Comorbidity Index; BMI, body mass index; CONUT score, Controlling Nutritional Status score; HbA1c, hemoglobin A1c; NGSP National Glycohemoglobin Standardization Program; Table 1A, Parameters of the patients in this study; Table 1B, Treatment the patients this study received; Table 1C, Nutrition and Glycemic Status of the patients this study at admission.

**Table 2 viruses-17-00643-t002:** Comparison of body composition between survivors and non-survivors.

Variables	All (*n *= 89)	Survivors (*n *= 57)	Non-Survivors (*n *= 32)	*p*-Value
Psoas muscle volume (cm^3^)	281.6 (201.3–405.6)	311.4 (231.8–417.2)	227.5 (182.1–298.7)	0.019
Combined pectoralis major and minor muscle areas (mm^2^)	2949 (2339–3837)	3304 (2507–4210)	2740 (2222–3329)	0.011
Erector spinae muscle area (mm^2^)	3046 (2437–3641)	3352 (2653–3893)	2467 (2102–3446)	0.001
Subcutaneous fat volume (cm^3^)	4362 (3254–5546)	4928 (3426–6681)	3735 (2759–4855)	0.003
Visceral fat volume (cm^3^)	4471 (3474–6227)	4475 (3512–6254)	4151 (3063–6149)	0.531

Nonparametric continuous variables are expressed as medians and interquartile ranges (first and third quartiles). The significance level was set at 5%. Abbreviations: Survivors, Survivors at the time of discharge from the ICU or transfer; Non-Survivors, Non-survivors at the time of discharge from the ICU or transfer.

**Table 3 viruses-17-00643-t003:** Comparison of daily rate of change in body composition between survivors and non-survivors.

Variable	Total (*n *= 30)	Survivors (*n *= 15)	Non-Survivors (*n *= 15)	*p*-Value
Change rate of pectoralis muscle area (%/day), median (IQR)	−0.246 (−1.355–0.150)	−0.434 (−1.355–−0.092)	−0.156 (−1.514–2.110)	0.064
Change rate of erector spinae area (%/day), median (IQR)	−0.631 (−1.364–−0.076)	−0.586 (−1.247–−0.115)	−1.007 (−1.672–0.392)	0.202
Change rate of psoas muscle volume (%/day), median (IQR)	−0.911 (−1.404–−0.075)	−0.826 (−1.022–−0.067)	−1.305 (−1.424–−0.200)	1.00
Change rate of subcutaneous fat volume (%/day), median (IQR)	−0.174 (−0.618–0.751)	−0.550 (−0.768–−0.236)	0.792 (−0.137–1.388)	0.043
Change rate of visceral fat volume (%/day), median (IQR)	−0.254 (−0.844–0.180)	−0.226 (−0.783–−0.057)	−0.403 (−0.990–−0.199)	0.829

Abbreviations: Survivors, Survivors at the time of discharge from the ICU or transfer; Non-Survivors, Non-survivors at the time of discharge from the ICU or transfer.

**Table 4 viruses-17-00643-t004:** Multivariate logistic regression analysis of factors associated with mortality.

Explanatory Variable	Odds Ratio	95% CI	*p*-Value
APACHE II score	0.834	0.741–0.938	0.002
Psoas muscle volume (cm^3^)			
Total area of the pectoralis major and minor muscles (mm^2^)			
Erector spinae muscle area (mm^2^)	1.001	1.000–1.002	0.031
BMI			
Blood glucose (mg/dL)			
D-dimer (μg/mL)			
Lactate (mmol/L)			
Use of mechanical ventilation			
Use of nafamostat			

Multivariate logistic regression analysis was conducted to identify factors associated with mortality. Odds ratios (ORs) and 95% confidence intervals (CIs) were calculated for each variable. Statistical significance was set at *p *< 0.05. Blank entries indicate variables that were not significant or not included in the final model. Abbreviations: APACHE II score, Acute Physiology, and Chronic Health Evaluation II score.

## Data Availability

Data supporting the findings of this study are available from the corresponding author, J.Y., upon reasonable request.

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
