# Peer review of "Changes in Body Composition During Intensive Care Unit Stay and Outcomes in Patients with Severe COVID-19 Pneumonia: A Retrospective Cohort Study"

_viruses, 2025, doi:10.3390/v17050643_

Round 1

Reviewer 1 Report

Comments and Suggestions for Authors

Dear Authors,

the article focused on the quantification by CT of muscle volume as predictor of mortality in severe Covid-19 pneumonia admitted in ICU.

The main result of your study was that poor ESM volume is a significant predictor of mortality, in association with Apache score II.

Your results are interesting, and represent a novelty. In addition, the article is well-written and clearly presented both in the methods and results sections.

Nevertheless, few minor issues should be addressed.

  1. could you better describe how did you perform the measurement of muscle volume at CT (maybe presenting a figure), for example if you use contrast-enhanced CT or not, which was the indication for the CT, etc.
  2. it is not completely clear how did you introduce variables in the multivriable analysis, could you please present a table of the univariable analysis results? It sounds strange that age was not included in the final model...

Reviewer 2 Report

Comments and Suggestions for Authors

This is a well written study on a topic of some interest that extends either ICU patients and COVID-19 patients. However, there are some methodological issues that I believe need to be addressed.

1) I am not entirely certain that COVID-19 itself is a waste disease as opposed to critical illness in general. No 8 in the reference used as citation for this claim does not fully support it. I am fine with the suggestion that covid-19 COULD be a wasting disease, but I do not feel this has been proven and certainly not by the given citation. I would adjust wording accordingly.

2) line 131, i feel you meant to say "exploratory" instead of "explanatory". Why did you choose the cutoff at 0.2 instead of (in my humbe opinion) the more usual and safe one of 0.1?

3) you need to define your primary endpoint better in methods. Are we talking about ICU survival, hospital survival, 30 day survival?

4) I don't think the numbers for gender (or other categorical variables) are useful to be given as x:y since you're already saying that you're reporting males and we can see the total number in each column's top with N.This is a personal style opinion, of course, there's nothing wrong with what you've done

5) some table lines seem misaligned. It's a formatting issue that possibly the journal editors can fix

6) It would be useful for the reader if apart from individual comorbidities that you select to present, if you have available Charlson index scores to present.

7) unless I've missed it, you need to report that the rate changes were calculated for the 30 individuals with multiple CT scans (and how many scans weere there?)

8) I do not understand the point of propensity score matching. You have balanced survivors and non-survivors around age and gender to eliminate them as confounders, but what I think you should have done instead, was include them in your multivariate model. PSM would have more sense if there was an intervention and you wanted to balance the intervention groups, instead of balancing the outcome groups. I also do not understand what you've done in your regression model. Did you try multiple models and keep the one you prefer? When you set up a decision in methods to include XYZ variables, changing this decision later after investigating, seems not methodologically sound to me.

9) in the PSM population, if you choose to keep your analysis as is, you should offer the bivariate logistic regression in a supplementary table at least.

Round 2

Reviewer 2 Report

Comments and Suggestions for Authors

I would like to thank the authors for addressing my concerns, with the PSM issue in particular, which I considered the main issue at hand. The addition of CCI data, in my opinion, also offers more information to the reader and adds value to the manuscript. Well done.

I believe that referring to "survival" as "ICU survival" instead of "hospital survival" is more appropriate, given that your follow-up would stop with transfers. 

With the improvements, I believe the masnucript is suitable for publication.
